# DIVERSITY–AMBIGUITY EXPLORATION FOR WEAKLY SUPERVISED VIDEO ANOMALY DETECTION

## ABSTRACT

Weakly supervised learning provides a cost-effective framework to video anomaly detection by using video-level supervision instead of relying on the costly fine-grained segment-level labels. Although contemporary methods have shown promising results on challenging real-world surveillance videos, most of them are evaluated using the Area Under the Receiver Operating Characteristic Curve (AUROC). Our work reveals that a high AUROC could result in a very low recall given a meaningful False Positive Rate (FPR) threshold. Thus, these models suffer from limited practical values, especially in high-stake domains (*e.g.,* public safety and medical diagnosis), where missing the true anomalies are highly costly. This surprising phenomenon is rooted in the interplay of weak supervision and the highly imbalanced distribution between normal and abnormal segments. To tackle this key challenge of building practical video anomaly detection systems, we propose a novel dual exploration strategy that combines temporal clustering with uncertainty-based segment exploration. Temporal clustering selects diverse segments based on both semantic and temporal similarity, while uncertainty-based sampling targets low-scoring segments with high model uncertainty. This dual exploration strategy ensures the model learns from a wide range of patterns, both diverse and ambiguous, resulting in more informed and robust decision-making, and reduction in false negatives. Meanwhile, we recommend two practical metrics to replace the commonly used AUROC score for a more effective measure for evaluation. Experiments conducted in challenging real-world videos demonstrate better dual exploration performance compared to competitive baselines on these metrics, which justifies its improved practical value in real-world settings.

## 1 INTRODUCTION

Video anomaly detection (VAD) aims to identify unusual events within video footage. One of its primary applications is in intelligent surveillance camera systems that assist public safety officers by automatically detecting abnormal activities. In these security-critical contexts, accurate anomaly detection is essential, as missed detections can lead to severe consequences such as delayed emergency responses, property damage, or threats to human life. Weakly supervised learning provides a practical and scalable solution for VAD by utilizing only on video-level labels, thereby avoiding the labor-intensive and costly process of segmentation-level annotation. As a representative weakly supervised method, multiple instance learning (MIL) treats each video as a bag of segments labeled either positive (containing at least one abnormal segment) or negative (containing only normal segments) (Dietterich et al., 1997). Features are extracted from video snippets using pretrained models such as C3D (Tran et al., 2015), I3D (Carreira & Zisserman, 2017), or ViT (Dosovitskiy et al., 2020), which are then used to train anomaly detectors that output anomaly scores. These detectors are usually trained to maximize the score margin between segments of the positive and negative bags, respectively.

State-of-the-art weakly supervised video anomaly detection (WSVAD) methods have achieved promising performance (Majhi et al., 2025; Wu et al., 2024; Chen et al., 2024a; Zhou et al., 2023; Lv et al., 2023). As a threshold-free metric, the Area Under the Receiver Operating Characteristic Curve (AUROC) score has been commonly used in the literature for performance evaluation. However, our detailed analysis reveals that AUROC often provides an overly optimistic assessment on the model's utility in practical settings. In its original form, AUROC does not account for the sig-

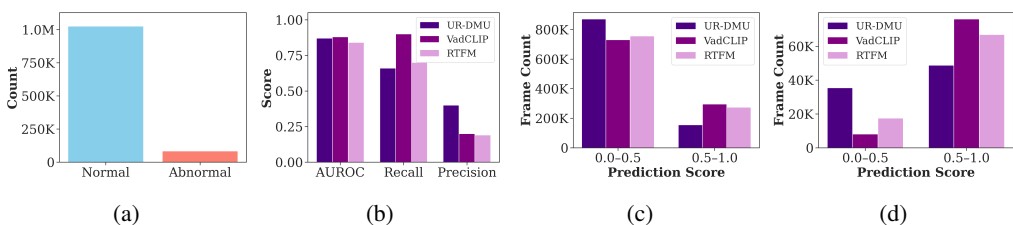

Figure 1: (a) Imbalanced distribution of normal and abnormal segments in UCF-Crime (b) AUROC, recall, and precision (at threshold 0.5) of three representative VAD models (c) & (d) Prediction scores of the models for normal and abnormal segments, respectively.

nificant difference in the cost of missing an abnormal event (false negatives) compared to a normal one (false positives) (Maurer & Pontil, 2020; Shao et al., 2023). For example, the UR-DMU model (Zhou et al., 2023) attains a competitive AUROC score of 86.97 on the UCF-Crime dataset (Sultani et al., 2018), but achieves the lowest recall compared to the other two models, including VadClip (Wu et al., 2024) and RTFM (Tian et al., 2021a), as shown in Figure 1 (b).

A fundamental reason for the low recall is the lack of segment-level supervision. To take advantage of (weak) video-level labels, existing works usually consider top-1 or $k$ segments with the highest anomaly scores as abnormal segments during training (Zadrozny & Elkan, 2002). However, a fixed $k$ value can either overestimate or underestimate the true number of abnormal events in different videos. Moreover, top-$k$ segments often cluster within a narrow temporal window, limiting the model's ability to capture multiple distinct abnormal events. Finally, this strategy tends to favor segments with prominent motion and makes the model biased toward simple contexts while overlooking more subtle or complex anomalies (Lv et al., 2023). Techniques, such as distributionally robust optimization (DRO) (Sapkota et al., 2021) and inclusion of temporally distant segments (Sapkota & Yu, 2022), have attempted to alleviate the problem. However, segment selection still relies on the highest anomaly scores, which are typically biased toward simpler contexts and miss out on complex patterns of anomalies, resulting in high false negatives.

Unbiased MIL mitigates context bias by splitting video segments into confident and ambiguous sets, using unsupervised clustering to detect anomalies in the ambiguous set (Lv et al., 2023). However, it assumes that all anomalies share similar semantics, limiting its ability to capture diverse patterns. It is also worth noting that making predictions biased towards abnormal segments does not address the problem either. For example, in Figure 1 (b), VadClip achieves a very high recall at 0.9 but it introduces more than 145K false positive, which almost double the size of UR-DMU, as shown Figure 1 (c). Figure 1 (d) further confirms that VadClip tends to predict higher anomaly scores. Due to the highly imbalanced distribution of normal and abnormal segments, it misclassifies a large number of normal segments, leading to a very high false positive rate that makes it less useful.

To address the challenges as outline above, we propose a dual diversity-ambiguity exploration strategy for WSVAD. It is designed to specifically target the discovery of all relevant abnormal segments, rather than focusing solely on those with the highest anomaly scores. It performs two complementary forms of exploration: (1) temporal clustering-based selection (TCBS), which selects semantically and temporally diverse segments to preserve local continuity and capture distinct abnormal events, and (2) uncertainty-based selection (UBS), which targets ambiguous segments with low anomaly scores but high model uncertainty. For uncertainty-guided exploration, we further design a memory unit that represents diverse abnormal events. Among the segments with high prediction uncertainty, we prioritize those that closely resemble the memory features for further exploration. Moreover, to mitigate the model's limited knowledge of truly novel or unseen anomalies, we leverage the extensive pretrained semantic knowledge of a vision-language model (VLM) and fuse its prediction scores with our dual-exploration strategy to improve overall detection performance. Our main contributions are summarized as follows:

- We identify a critical weakness in using AUROC for performance evaluation of video anomaly detection by showing its overly optimistic assessment on the model's utility in practical settings.
- We propose a dual exploration framework that leverages the temporal-semantic similarity of segments and model uncertainty under weak supervision to perform systematic exploration, enabling the discovery of diverse and ambiguous abnormal patterns.

- We design a memory unit to store the diverse patterns in the abnormal events and use them to guide exploration to avoid choosing random noise.
- We recommend two practical evaluation metrics to replace the commonly used AUROC score for a more effective measure for evaluation.

Experiments conducted in challenging real-world videos demonstrate better dual exploration performance compared to competitive baselines on these metrics, which justifies its improved practical value in real-world settings.

## 2 RELATED WORK

Sultani et al. proposed a deep MIL framework and a large-scale dataset for video anomaly detection (Sultani et al., 2018). The model is trained to maximize the separation of the highest-scoring segment in each video. The reliance on a single segment makes it vulnerable to outliers (Sapkota & Yu, 2022). Robust Temporal Feature Magnitude Learning (RTFM) considers top-$k$ abnormal segments for learning (Tian et al., 2021a).A few other recent methods also adopt the top-$k$ approach for segment selection with improved performance (Zhou et al., 2023; Wu et al., 2024; Chen et al., 2024a). However, they still suffer from some key limitations. First, using a fixed $k$ is suboptimal, as the actual number of abnormal segments in a video is typically unknown. Second, these methods often tend to select temporally adjacent high-scoring segments, potentially overlooking anomalies that are temporally dispersed. Finally, they tend to only capture confident, easy-to-classify segments, which might miss more complex and subtle anomalies.

To overcome these limitations, a Distributionally Robust Optimization (DRO) framework has been developed that adapts $k$ according to the content of each video (Sapkota et al., 2021). However, this approach still does not ensure the coverage of all distinct abnormal events. Dynamic non-parametric clustering has also been leveraged that encourages temporally adjacent segments to lie within the same cluster (Sapkota & Yu, 2022). Then, the segment with the highest anomaly score is selected from each cluster. However, this model still favors simple contextual anomalies and may miss complex or subtle patterns during segment selection and model training. In contrast, Unbiased MIL divides video segments into confident and ambiguous sets based on prediction variance, and clusters the ambiguous set into presumably normal and abnormal segments (Lv et al., 2023). Segments similar to high-scoring confident segments are labeled abnormal and considered for training. However, it assumes semantic similarity among anomalies. Diverse or distinct abnormal events may be missed, limiting exploration of the full anomaly spectrum, thereby compromising detection performance.

In parallel, increasing attention has been paid to the integration of textual features of CLIP-based models and the use of learnable prompts to enhance semantic diversity (Wu et al., 2024; Chen et al., 2024a; Joo et al., 2023). For instance, Zhong et al. (Wu et al., 2024) proposed VADClip, which combines visual and textual modalities by leveraging the frozen CLIP backbone for improved anomaly localization and classification. Although CLIP-based models and memory-augmented architectures (Zhou et al., 2023) offer rich and discriminative feature representations, these methods still depend on top-$k$ selection, limiting their ability to detect a diverse set of abnormal events. Beyond CLIP, recent vision-language models such as InternVL (Wang et al., 2025; Chen et al., 2024b), LLaVA (Li et al., 2024), and VideoLLaMA (Zhang et al., 2025) have demonstrated strong performance on general video understanding tasks. While strategies such as learning reasoning modules, guided prompts (Ye et al., 2025), or instruction tuning Zhang et al. (2024) can improve their performance on VAD task, these approaches incur additional costs for fine-tuning. Instead, we leverage the confident knowledge of these VLMs in a training-free manner to enhance detection performance.

## 3 DIVERSITY-AMBIGUITY DUAL EXPLORATION FOR WSVAD

### 3.1 PRELIMINARIES

In a MIL setting, each abnormal video is treated as a positive bag $\mathcal{B}_{\text{pos}}$, and has at least one abnormal segment, whereas each normal video is treated as a negative bag $\mathcal{B}_{\text{neg}}$, containing only normal segments. During training, the model is guided by video-level labels and learns to assign an anomaly score at the segment level. The model $f(.)$ is encouraged to assign high anomaly scores to abnormal

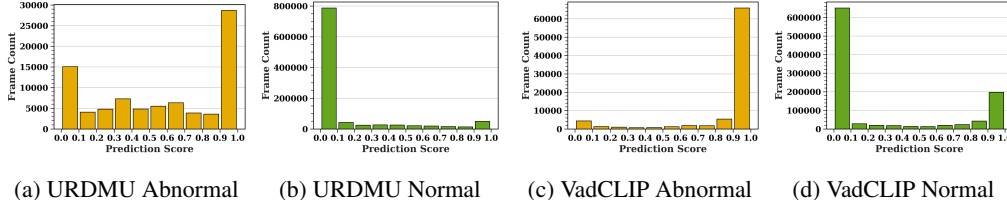

(a) URDMU Abnormal    (b) URDMU Normal    (c) VadCLIP Abnormal    (d) VadCLIP Normal

Figure 2: Prediction score distributions from two WSVAD methods, highlighting AUROC limitations. (a, b) UR-DMU misses 15k abnormal segments (low recall) but classifies nearly all normal segments correctly. (c, d) VadCLIP detects more abnormal segments, improving recall, but misclassifies 200k+ normal segments at a 0.9 threshold. Severe imbalance makes AUROC deceptively high.

segments (close to 1) and low scores (close to 0) to normal ones. Then the MIL loss is defined as:

$$\mathcal{L}_{MIL} = \text{BCE}(y, \hat{y}) \tag{1}$$

where $y = [y_{\text{pos}} = 1, \ y_{\text{neg}} = 0]$ is the ground truth bag level. For most existing methods, the positive bag label is determined by either selecting the maximum scoring segment or by averaging the scores of top-$k$ segments in the positive bag. Following the MIL assumption, let $\mathcal{K} \subset \mathcal{B}_{\text{pos}}$ be the set of selected segments from the positive bag such that:

$$\hat{y} = \left[ \hat{y}_{\text{pos}} = \frac{1}{|\mathcal{K}|} \sum_{k \in \mathcal{K}} f(\mathbf{x}_{[k]}^+), \ \hat{y}_{\text{neg}} = \max_{j \in \mathcal{B}_{\text{neg}}} f(\mathbf{x}_j^-) \right] \tag{2}$$

where $\mathbf{x}_i$ denotes the feature of the $i$-th video segment with superscript $^+$ or $^-$ indicating that the segment belongs to the positive and negative bags, respectively. In a negative bag, since all video segments are normal, minimizing the maximum anomaly score encourages the model to assign low scores across the entire video.

### 3.2 Why is AUROC not a good metric for Evaluating Video Anomaly Detection?

In cases of severe data imbalance where negatives vastly outnumber positives, even a very low false positive rate (FPR) may still generate a large number of false alarms. To maintain precision at a usable level, the classification threshold must be raised, which in turn compromises recall. We refer to this as a **low practical recall phenomenon**. Meanwhile, since AUROC measures the probability that a model ranks a randomly chosen positive instance above a negative one, it can still appear high if the model correctly identify many negatives (which are more frequent) at a good rate given the class imbalance setting. Thus, AUROC gives a false sense of effectiveness in such scenarios, masking the severe recall loss required to maintain precision.

This limitation becomes apparent in the preliminary analysis of the WSVAD models. For instance, UR-DMU fails to detect about 15,000 abnormal segments even at a low threshold, yet maintains a high AUROC score of 0.8697 because it correctly classifies a large portion of normal segments [see Figures 2a and 2b] . Conversely, models like VadClip that leverages large pre-trained vision–language encoders such as CLIP achieve better recall at the expense of excessive false alarms. In the Figure 2d, VadClip misclassifies roughly $2 \times 10^5$ normal segments in the 0.9–1.0 prediction score bin, which inflates AUROC, as the false positive rate remains small relative to millions of normal segments. However, in a real-world surveillance setting, this is not a reliable solution, as a security officer needs to review tens and thousands of false alarms. Ultimately, these models exhibit a low practical recall at effective operating thresholds (see the results at Table 1).

The cost-sensitivity limitation of AUROC has been alleviated via weighted optimization methods based on AUC Shao et al. (2023) or partial AUC (pAUC) Maurer & Pontil (2020) have been formulated. However, they are not suitable for WSVAD. Considering the significantly higher cost of false negatives in this context, we argue that recall evaluated at a meaningful FPR threshold, along with Average Precision, provides a more practical and effective evaluation metric.

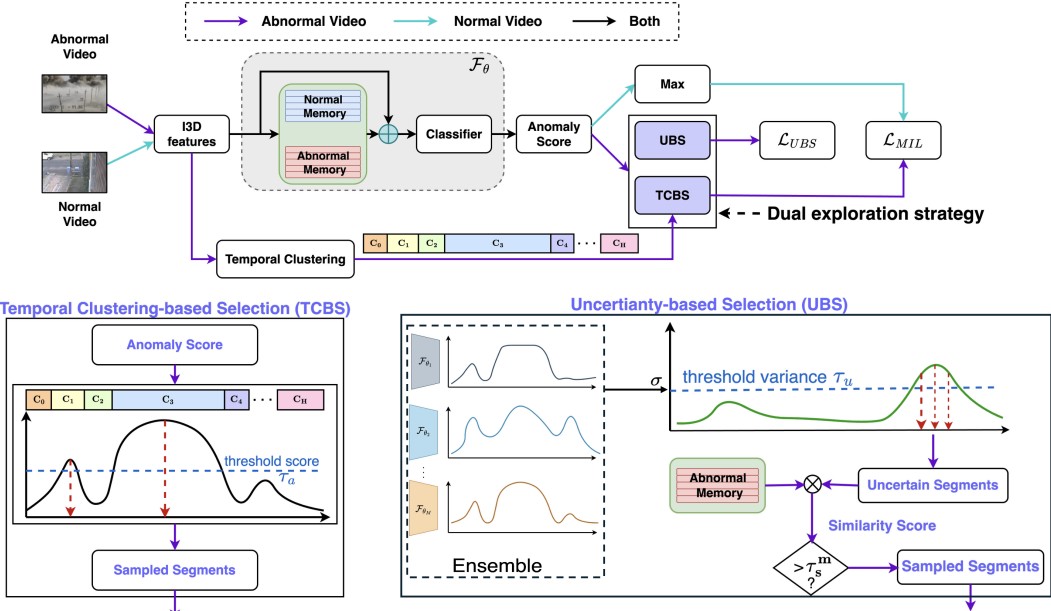

Figure 3: Overview of the proposed MIL framework with a dual exploration strategy. After the model generates segment-level anomaly scores, we apply segment exploration and selection for computing the MIL loss. For normal videos, the segment with the highest anomaly score is selected to compute the MIL loss. For abnormal videos, segments are selected using a dual exploration strategy: (1) Temporal clustering-based selection, which chooses top-scoring segments from distinct temporal clusters to ensure diversity among selected events; and (2) Uncertainty-based selection, which targets segments with high predictive uncertainty and strong similarity to the abnormal memory, encouraging exploration of ambiguous yet informative patterns.

## 3.3 OVERVIEW OF THE DUAL EXPLORATION FRAMEWORK

The conventional approach in WSVAD is to extract segment-level features from pretrained backbones. Although recent works increasingly prefer the CLIP model, as pointed out in the observation section, these methods tend to produce significant false alarms. Following the common practices and memory augmented baselines (Zhou et al., 2023), we extract features from the I3D model pretrained on the Kinetics dataset, and pass them to the Global-Local Temporal Module and Memory Units (see Figure 3). The classifier then generates segment-level anomaly scores, which are used to compute MIL Loss. Most previous approaches then select only the top-$k$ segments with the highest anomaly scores, restricting the model's ability to consider all segments in a positive bag, potentially overlooking other abnormal segments, and limiting its understanding of the full range of abnormalities present. To mitigate this bias, a broader exploration of segments within positive bags is essential. If we allow the model to observe a wider range of segments during training, it can make much more informed decisions. However, we cannot simply perform explicit exploration strategies such as reinforcement learning due to the lack of fine-grained supervision. Therefore, we propose a soft exploration strategy that combines the selection of temporally diverse and ambiguous segments. The key idea is to enable the model to consider all potentially abnormal segments and make decisions based on a comprehensive view of the video.

## 3.4 TEMPORAL CLUSTERING BASED SELECTION (TCBS)

We adopt a clustering-based approach to encourage semantic diversity among selected segments. Video segments are clustered based on both feature and temporal similarity, ensuring that each cluster contains visually similar segments that occur closer in time. This encourages each cluster to represent a single scene while separating segments from different scenes into distinct clusters. During selection, we allow at most one segment to be chosen from any given cluster, promoting diversity across scenes without overrepresenting any single one.

Let $\mathbf{x}_i$ denote the feature representation of the $i^{th}$ segment. We define a similarity function $S(\mathbf{x}_i, \mathbf{x}_j)$ as the cosine similarity between segments. We perform temporal clustering as outlined in Algorithm 1. Specifically, the first segment initializes the first cluster. For each subsequent segment, if its similarity to the representative feature of the previous cluster ($\mathbf{x}_m^c$) exceeds the predefined threshold $\tau_s$, it is added to that cluster; otherwise, a new cluster is created.

$$S(\mathbf{x}_i, \mathbf{x}_j) = \frac{\mathbf{x}_i^\top \mathbf{x}_j}{\|\mathbf{x}_i\|\|\mathbf{x}_j\|} \tag{3}$$

We set the threshold $\tau_s$ based on the cosine similarity between adjacent segments within each video. For videos with drastic motion or scene changes, like explosions, the similarity between consecutive segments tends to be lower compared to more subtle anomaly scenarios, such as abuse or shoplifting. To make the threshold adaptive, we compute $\tau_s$ for each video individually by selecting the $q$-quantile (e.g., $q = 0.03$) of its similarity scores. The hyperparameter $q$ controls the number of clusters formed. Figure 8a illustrates the distributions of clusters formed in training videos with varying $q$ values. For example, setting $q$ to the 3rd percentile (i.e., $q = 0.03$) results in most videos having around 10 clusters. Lowering $q$ leads to fewer clusters, since segments must exhibit greater dissimilarity to initiate a new cluster. Figure 8b shows the distribution of the similarity threshold $\tau_s$ across both datasets. As $q$ represents the quantile of the similarity distribution, increasing $q$ naturally leads to a higher similarity threshold. The UCF-Crime dataset has generally higher similarity thresholds because it consists surveillance videos with visually similar segments, whereas the XD-Violence dataset contains movie clips with varied camera angles and cuts. Despite these differences, our adaptive thresholding approach works effectively across both datasets.

To ensure selection from diverse abnormal events, we choose the segment with the highest anomaly score from each cluster. Let $f(\mathbf{x}_i^+)$ be the anomaly score of the $i^{th}$ segment. A segment is included in the set $\mathcal{K}$ only if its anomaly score exceeds a predefined threshold $\tau_a$.

$$\mathcal{K} = \bigcup_{i=1}^{H} \{k : k = \arg\max_{j \in C_i} p_j, \quad f(\mathbf{x}_k^+) \geq \tau_a\} \tag{4}$$

where $H$ is the total number of clusters, $k$ represents the segment with the highest anomaly score within cluster $C_i$, and $\tau_a$ is a dynamic threshold, which is chosen as a percentile of the anomaly scores computed over the segments of a given video.

TCBS ensures that segments representing abnormal events at different times are selected, rather than consecutively ranked segments that may belong to a single event.

### 3.5 UNCERTAINTY-BASED SELECTION

As shown in Figure 9, we observe segments in abnormal video with low anomaly score that may not be captured by temporal clustering alone, as it relies on certain anomaly score thresholds for detection. However, we find that these segments often exhibit high prediction uncertainty. Uncertainty-based exploration encourages the model to focus on such segments during the early training stages, allowing it to develop a robust understanding of abnormal patterns by prioritizing regions with high predictive uncertainty. Such exploration prevents the model from overfitting on highly confident segments, ensuring it becomes well-informed about its ambiguities.

Segment-level uncertainty $u_i$ is obtained by calculating the standard deviations of the anomaly score obtained from an ensemble of $M$ models $f(.)$ trained with different initialization. To differentiate abnormal segments from normal ones among the uncertain cases, we select the segments that closely resemble abnormal memory. We then define a set of candidate segments $\mathcal{U}$ that satisfy both criteria: (i) high uncertainty ($u_i > \tau_u$) and (ii) high semantic similarity to abnormal memory ($\frac{1}{|AM|} \sum_j S(\mathbf{x}_i^+, \mathbf{x}_j^{\mathbf{AM}}) > \tau_s^m$), where $j$ indicates the memory slot, and $|AM|$ is the size of abnormal memory (refer to Algorithm 2). For segments in $\mathcal{U}$, we compute the BCE loss. Since uncertainty-based selection is applied only to abnormal videos, the target score for these segments is set to 1.

$$\mathcal{L}_{UNC} = \frac{1}{|\mathcal{U}|} \sum_{i \in \mathcal{U}} \text{BCE}(f(\mathbf{x}_k^+), 1), \tag{5}$$

Uncertainty is calculated at each epoch and updated during the training. As the training starts with random initialization, the initial uncertainty estimates are unreliable. It is reasonable to use uncertainty estimates only after the model has undergone a few training steps. On the other hand,

we aim to avoid overemphasizing uncertain segments, as they might overfit to noisy or borderline samples. Excessive focus on such segments can mislead the model and degrade it's generalization performance.

A good approach is to gradually reduce the influence of uncertainty as training progresses like implementing a cosine annealing scheduler. While we do not explicitly implement such a scheduler, we find that by carefully setting the hyperparameters $\tau_s^m$ (memory similarity threshold) and $\tau_u$ (uncertainty threshold), we can naturally obtain a similar scheduling effect. In the early training phase, the memory is randomly initialized, and the cosine similarity between memory entries and uncertain input segments is generally below 0.7. After a few training steps, as the model begins to learn abnormal patterns, the similarity of potential abnormal segments with abnormal memory starts to increase. Furthermore, by selecting uncertain but highly similar segments across all ensemble models and including them in training, the overall model uncertainty decreases over time.

Finally, we integrate MIL loss ($\mathcal{L}_{\texttt{MIL}}$) and uncertainty based selection loss ($\mathcal{L}_{\texttt{UNC}}$) along with the baseline model loss ($\mathcal{L}_{\texttt{URDMU}}$):

$$\mathcal{L}_{\texttt{Dual}} = \mathcal{L}_{\texttt{MIL}} + \gamma \mathcal{L}_{\texttt{UNC}} + \mathcal{L}_{\texttt{URDMU}} \tag{6}$$

### 3.6 Knowledge Fusion with VLM

In our approach, we primarily rely on abnormal memory for the exploration of uncertain samples, which is biased towards the model's prior knowledge of abnormality. As a result, truly novel or unseen anomalies that differ significantly from the memory unit may be overlooked. To address this limitation, we propose to leverage the generalization ability of powerful VLMs by fusing their prediction scores with the dual exploration model.

Let $y_{\text{Dual}}$ denote the prediction from the Dual Exploration Model trained with the loss $\mathcal{L}_{Dual}$, and $y_{\text{VLM}}$ denote the prediction from the Vision-Language Model (VLM) pretrained on a large-scale dataset. $y_{\text{VLM}}$ represents the VLM's confidence computed as the average of output across multiple runs using training-free inference with a structured prompt (see Appendix). The combined prediction $y_{\text{combined}}$ is obtained as a weighted sum of the two predictions, where the weight $\lambda \in [0, 1]$ is tuned for optimal performance.

$$y_{\text{combined}} = \lambda \, y_{\text{Dual}} + (1 - \lambda) \, y_{\text{VLM}} \tag{7}$$

## 4 Experiments

**Datasets.** We evaluate our model on two publicly available WSVAD benchmark datasets. The **UCF-Crime** dataset (Sultani et al., 2018) consists of unedited surveillance footage covering 13 types of abnormal events. It contains 1,900 videos, in which the training split has 1,610 videos with video-level annotations, and the test set has 290 videos with segment-level annotations. The **XD-Violence** dataset (Wu et al., 2020) includes 4,754 untrimmed videos with only video-level labels, consisting of diverse content captured from movies and YouTube.

**Evaluation metrics.** The standard evaluation across all prior works uses AUROC for UCF-Crime and Average Precision (AP) for XD-Violence (Wu et al., 2020; Sultani et al., 2018). However, as discussed earlier, AUROC tends to become overly optimistic and is not a reliable metric for safety-critical scenarios. Therefore, we evaluate both datasets using the **Area Under the Precision-Recall Curve**, and **Recall at a False Positive Rate (Recall@FPR)**. Recall@FPR $= \alpha$ measures the proportion of true anomalies correctly detected when the false positive rate is constrained to a fixed threshold $\alpha$. For instance, Recall@FPR $= 1\%$ quantifies the recall achieved while keeping the false alarm rate below 1%. Unlike AUROC, which averages performance across all thresholds, including those corresponding to unrealistically high false positive rates, Recall@FPR provides a targeted measure of detection capability under practically feasible false alarm levels, making it more suitable for real-world deployment. We also compare the misclassification cost for the baseline URDMU and our exploration method by assigning a cost $W$ of missing abnormal segments (Bishop, 1995).

$$\texttt{MCC} = W \times \texttt{FN} + \texttt{FP} \tag{8}$$

where $\texttt{FN}$ and $\texttt{FP}$ are the count of false negatives and false positives, respectively, and $W$ is the penalty assigned to false negatives.

Details on implementation, hyperparameters, and experimental setup are included in the Appendix.

Table 1: Results on UCF-Crime dataset. * indicates our own implementation

| Method | Feature | AP (%) | Recall@FPR=1% | Recall@FPR=2% | Recall@FPR=3% |
|---|---|---|---|---|---|
| MIL (Sultani et al., 2018) | I3D | 25.03* | - | - | - |
| RTFM (Tian et al., 2021a) | I3D | 29.46* | - | - | - |
| BNSVP (Sapkota & Yu, 2022) | I3D | 30.68* | - | - | - |
| MGFN(Chen et al., 2023) | I3D | 18.88* | - | - | - |
| UR-DMU (Zhou et al., 2023) | I3D | 35.48* | 0.170 | 0.170 | 0.212 |
| UR-DMU (various $k$): | | | | | |
| $\quad k = 1$ | I3D | 33.94* | - | - | - |
| $\quad k = 5$ | I3D | 33.54* | - | - | - |
| PEL4VAD Pu et al. (2024) | I3D+Text | 33.99 | - | - | - |
| VadClip (Wu et al., 2024) | CLIP | 33.55 | 0.109 | 0.155 | 0.217 |
| InternVL3-14B (Chen et al., 2024b) | Training Free | 29.50 | 0.301 | 0.301 | 0.418 |
| Ours (Dual Exploration + InternVL) | I3D + InternVL | **40.75** | 0.173 | 0.263 | 0.336 |

Table 2: Results on XD-Violence dataset. * indicates our own implementation

| Method | Feature | AP (%) | Recall@FPR=4% | Recall@FPR=5% | Recall@FPR=6% |
|---|---|---|---|---|---|
| MIL (Sultani et al., 2018) | C3D | 75.68 | - | - | - |
| RTFM (Tian et al., 2021a) | I3D | 77.81 | - | - | - |
| CRFD (Wu & Liu, 2021) | I3D | 75.90 | - | - | - |
| MSL (Wei et al., 2022) | V-Swim | 78.28 | - | - | - |
| UR-DMU (Zhou et al., 2023) | I3D | 79.14* | 0.590 | 0.638 | 0.682 |
| TSA Joo et al. (2023) | CLIP | 82.17 | - | - | - |
| MACIL-SD Yu et al. (2022) | I3D+audio | 83.40 | - | - | - |
| TPWNG Yang et al. (2024) | CLIP | 83.68 | - | - | - |
| VadCLIP Wu et al. (2024) | CLIP | 84.50 | 0.635 | 0.672 | 0.701 |
| InternVL3-14B (Chen et al., 2024b) | Training Free | 69.85 | 0.637 | 0.637 | 0.755 |
| Ours (Dual Exploration + InternVL) | I3D + InternVL | **84.58** | 0.657 | 0.715 | 0.752 |

## 4.1 COMPARISON RESULTS

We report AP and Recall@FPR for UCF-Crime and XD-Violence in Tables 1 and 2. Since most UCF-Crime baselines report AUROC, we re-computed their AP scores using available code and models. Our method, combining dual exploration and VLM fusion, achieves the highest AP (40.75%) among both weakly supervised models and training-free VLMs. This improvement stems from temporal clustering and uncertainty-based exploration, in contrast to baselines that rely only on top-$k$ highest scoring segments. At 5% FPR on the UCF-Crime dataset model produces over 50,000 false alarms ($\approx 50$ extra videos), making such thresholds impractical. We therefore also report recall at lower FPRs, where our method outperforms other weakly supervised approaches. The training-free InternVL model shows stronger recall at very low FPRs, making it valuable for fusion despite its lower AP score. Its higher recall at low FPR indicates that we can rely on its confident predictions, but the model confuses many less obvious anomalies with normal frames, leading to poor overall ranking and increased false positives at lower thresholds. Similarly, for XD-Violence, our method consistently achieves higher recall than baselines, even at FPRs of 4% and above.

## 4.2 ABLATION STUDY

**Effectiveness of Dual Exploration**: Table 3 shows the effect of individual exploration components. Temporal clustering-based exploration achieves the highest recall at 1% FPR because it selects segments from different clusters rather than consecutive frames, ensuring coverage of diverse parts of the video. By limiting selection to one segment per cluster, it also reduces the likelihood of multiple false alarms from the same region. In contrast, uncertainty-based exploration alone yields lower recall at very low FPR, as it may select noisy normal segments that are semantically similar to anomalies. However, as FPR increases to 2–3% within the practical operating range, it captures segments that temporal clustering (TCBS) misses, specifically, segments with lower anomaly scores but high model uncertainty. By combining both strategies, our method achieves better practical recall, as shown by improvements at 2% and 3% FPR, and better AP despite a few noisy selections. Figure 4 demonstrates improved recall of dual exploration across all prediction thresholds, except

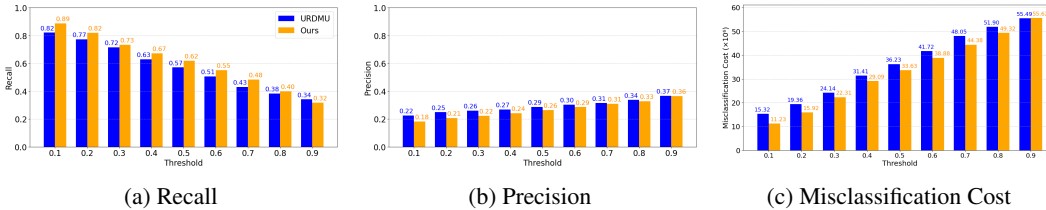

|                     | (a) Recall                  | (b) Precision               | (c) Misclassification Cost  |

Figure 4: Comparison of recall, precision, and misclassification cost with the baseline (UR-DMU) across different thresholds on the UCF-Crime dataset.

at 0.9 due to TCBS, with minimal loss in precision and a lower misclassification cost ($W = 1000$) compared to the baseline model.

Table 3: Ablation of different components on the UCF-Crime dataset.

| Baseline | TCBS | UBS | VLM | AP | Recall@FPR=1% | Recall@FPR=2% | Recall@FPR=3% |
|----------|------|-----|-----|------|---------------|---------------|---------------|
| ✓ |  |  |  | 35.48 | 0.170 | 0.170 | 0.212 |
| ✓ | ✓ |  |  | 34.25 | **0.182** | 0.182 | 0.207 |
| ✓ |  | ✓ |  | 34.58 | 0.146 | 0.192 | 0.243 |
| ✓ | ✓ | ✓ |  | 36.42 | 0.167 | 0.232 | 0.256 |
| ✓ | ✓ | ✓ | ✓ | **40.75** | 0.173 | **0.263** | **0.336** |

Table 4: AUROC per event type for Dual Exploration and VLM on UCF-Crime.

| Method | AUROC (Abnormal) | Abuse | Arrest | Arson | Assault | Burglary | Explosion |
|--------|------------------|-------|--------|-------|---------|----------|-----------|
| Dual Exploration | 0.708 | 0.81 | 0.64 | 0.60 | 0.89 | 0.77 | 0.47 |
| InternVL | 0.700 | 0.61 | **0.70** | 0.58 | 0.92 | 0.64 | **0.77** |

| Method | Fighting | Accidents | Robbery | Shooting | Shoplifting | Stealing | Vandalism |
|--------|----------|-----------|---------|----------|-------------|----------|-----------|
| Dual Exploration | 0.79 | 0.68 | 0.83 | 0.77 | 0.66 | 0.85 | 0.90 |
| InternVL | 0.76 | **0.82** | 0.81 | 0.74 | 0.52 | 0.84 | 0.76 |

**Effectiveness of VLM Fusion**: To understand the effectiveness of VLM fusion, we report AUROC separately for abnormal videos and for different types of abnormal events using dual exploration prediction and VLM prediction. In Table 4, we observe that VLM shows stronger detection capability for certain event types, such as Explosion, Arrest, and Road Accidents, but performs poorly on human-centric abnormal events that occur within short temporal windows (e.g., Abuse and Shooting) as well as events that require temporal reasoning (e.g., Shoplifting). This limitation may arise from the lack of reasoning ability in VLMs trained on generalized domains. In contrast, our dual exploration method achieves comparatively better performance on these challenging categories. Therefore, combining dual exploration with VLM enables more effective detection by leveraging knowledge sharing between generalized anomalies and domain-specific anomalies.

## 5 CONCLUSION AND FUTURE WORK

In this work, we emphasize the need for cost-sensitive evaluation metrics for WSVAD, as commonly used metrics like AUROC can either mask poor recall or obscure significant false alarm rates, leading to a low practical recall that limits real-world applicability. To improve recall, we introduce a soft exploration strategy that informs the model about potentially diverse and ambiguous abnormal events during the training process, rather than being restricted to top-$k$ segments. Additionally, we introduce a fusion approach that leverages knowledge sharing between vision-language models (VLMs) trained on generalized domains and exploration models trained on specialized domains to mitigate the exploration model's bias toward its learned abnormal features during uncertainty exploration. In the future, we plan to extend this knowledge sharing to the training phase, enabling both VLMs and weakly supervised models to improve their detection capabilities.

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

# Appendix

## A LIST OF SYMBOLS

Table 5: List of symbols and their meanings

| Symbol | Meaning |
|---|---|
| $q$ | Quantile threshold for similarity distribution |
| $\tau_s$ | Threshold for adjacent frame similarity |
| $\tau_a$ | Quantile threshold for anomaly score |
| $\tau_u$ | Threshold for uncertainty |
| $\tau_s^m$ | Threshold for memory similarity |
| $AM$ | Abnormal Memory |
| $\gamma$ | Weighting parameter in uncertainty loss |
| $\lambda$ | Weighting parameter in knowledge fusion |
| $\alpha$ | Desired False Positive Rate |
| $MCC$ | Misclassification Cost |
| $W$ | Weight to False Negatives in isclassification Cost |
| $S$ | Cosine similarity between two features |
| $N$ | Number of segments |
| $C$ | Cluster assignment of each segment in a video |
| $p_j$ | Anomaly score of segment $j$ |
| $\mathcal{K}$ | Set of selected segments for MIL loss |
| $\mathcal{U}$ | Uncertainty set |
| $AP$ | Average Precision |
| $AUROC$ | Area Under ROC Curve |
| $AUROC(Abnormal)$ | AUROC of only abnormal videos |

## B ALGORITHMS

In this section, we present pseudocode for Temporal Clustering-based selection 1 and Uncertainty-based exploration 2.

---

**Algorithm 1** Temporal Clustering of Video Segments

**Require:** Segment features $\{\mathbf{x}_1^+, \ldots, \mathbf{x}_N^+\}$, similarity threshold $\tau_s$.
1: Initialize set of clusters $\mathcal{C} \leftarrow \emptyset$
2: Create first cluster $C_1 \leftarrow \{1\}$, $\mathcal{C} \leftarrow \mathcal{C} \cup \{C_1\}$, set $m \leftarrow 1$
3: **for** $i = 2$ to $N$ **do**
4:    Compute similarity $S(\mathbf{x}_i^+, \mathbf{x}_m^c)$
5:    **if** $S(\mathbf{x}_i^+, \mathbf{x}_m^c) \geq \tau_s$ **then**
6:       Assign $i$ to $C_m$: $C_m \leftarrow C_m \cup \{i\}$
7:    **else**
8:       Create new cluster: $C_{m+1} \leftarrow \{i\}$, $\mathcal{C} \leftarrow \mathcal{C} \cup \{C_{m+1}\}$
9:       Update $m \leftarrow m + 1$
10:    **end if**
11: **end for**
12: **return** $\mathcal{C}$

---

**Algorithm 2** Uncertainty-Based Exploration of Video Segments

**Require:** $\{\mathbf{x}_1, \ldots, \mathbf{x}_N\}$, $\{u_1, \ldots, u_N\}$, $\tau_u$, $\tau_s^m$, abnormal memory $AM$
**Ensure:** A Set of selected uncertain segments $\mathcal{U}$
1: Initialize $\mathcal{U} \leftarrow \emptyset$
2: **for** $i = 1$ to $N$ **do**
3:    Compute:
   $s_i = \frac{1}{|AM|} \sum_{j \in AM} S(\mathbf{x}_i^+, \mathbf{x}_j^{AM})$
4:    **if** $u_i > \tau_u$ **and** $s_i > \tau_s^m$ **then**
5:       Add $i$ to $\mathcal{U}$: $\mathcal{U} \leftarrow \mathcal{U} \cup \{i\}$
6:    **end if**
7: **end for**
8: **return** $\mathcal{U}$
9:
10:
11:

## C  IMPLEMENTATION DETAILS

The hyperparameters $q$ and $\tau_u$ are fine-tuned over the ranges $[0.01,\ 0.1]$ (with a step size of 0.01) and $[0.1,\ 0.4]$ (with a step size of 0.05), respectively. We performed a grid search for the fusion weight parameter $\lambda$ over the interval $[0, 1]$ to identify the value that maximizes the AP score. The best-performing hyperparameters for the UCF-Crime dataset are: $q = 0.03$, $\tau_a = 0.955$, $\tau_u = 0.2$, $\tau_s^m = 0.7$, $\gamma = 1$, and $\lambda = 0.45$, While we can also set the same hyperparameter setting for XD-Violence, we get the best performance on $q = 0.03$, $\tau_a = 0.95$, $\tau_u = 0.3$, $\tau_s^m = 0.7$, $\gamma = 0.1$, and $\lambda = 0.51$ All other features, training settings, and baseline model hyperparameters are the same as in the UR-DMU model (Zhou et al., 2023), for example, a batch size of 64, a learning rate of 0.001, 60 memory slots, and so on.

For training-free inference with the InternVL3-14B model, we experiment with temporal windows of 48, 144, and 288 frames, and select a 48-frame window to capture short-clip anomalies, passing 8 frames at a time for inference. To estimate the model's confidence, we average the results over 5 runs, which is denoted by $y_{VLM}$.

For uncertainty calculation in the dual exploration, three models are trained. Although they can be run in parallel on three GPUs, we execute them sequentially on a single NVIDIA RTX A6000 with 50 GB of memory. With reduced testing frequency, evaluating every 100 steps instead of every 10, training takes only about one and a half hours for the UCF-Crime dataset.

## D  BASELINE COMPARISION

Since AUROC is the primary evaluation metric for the UCF-Crime dataset, Table 6 presents a comparison of AUROC and AP scores. As baseline models do not report AP for UCF-Crime, we re-trained them to evaluate this metric. Among single-modality models, URDMU achieves the highest AUROC, while for multi-modality models, VadCLIP shows the best performance. We have omitted recent baselines such as (Majhi et al., 2025) because neither their code nor their model is publicly available. This prevents us from computing their AP score, therefore, only best best-performing, open-source models are reported in Table 1.

Table 6: Results on UCF-Crime dataset including AUROC Score

| Method | Feature | AUROC (%) | AP (%) |
|---|---|---|---|
| MIL (Sultani et al., 2018) | I3D | 76.21 | 25.03* |
| RTFM (Tian et al., 2021b) | I3D | 84.30 | 29.46* |
| BNSVP (Sapkota & Yu, 2022) | I3D | 83.39 | 30.68* |
| MGFN(Chen et al., 2023) | I3D | 80.21* | 18.88* |
| UR-DMU (Zhou et al., 2023) | I3D | 86.97 | 35.48* |
| UMIL (Lv et al., 2023) | CLIP | 86.75 | – |
| TSA(Joo et al., 2023) | CLIP | 87.58 | – |
| TPWNG (Yang et al., 2024) | CLIP | 87.79 | – |
| PEMIL(Chen et al., 2024a) | I3D+Text | 86.83 | – |
| PEL4VAD Pu et al. (2024) | I3D+Text | 86.76 | 33.99 |
| VadClip (Zhou et al., 2023) | CLIP | 88.02 | 33.55 |
| InternVL3-14B (Chen et al., 2024b) | Training Free | 79.61 | 29.50 |
| Ours (Dual Exploration) | I3D | 85.63 | **36.42** |
| Ours (Dual Exploration + InternVL) | I3D + InternVL | 87.80 | **40.75** |

# E  EFFECT OF HYPERPARAMETERS

## E.1  EFFECT OF $\tau_a$

We study the effect of varying the threshold $\tau_a$ for the anomaly score in Table 8 and 7. For both datasets, we find that a threshold of 0.99 is too high for the model and misses a significant ratio of abnormal events, indicated by the low recall values. As we decrease the threshold, we find that around $\tau_a = 0.95/0.96$ results in a good balance between precision and recall. A lower threshold would increase recall but compromise precision.

Table 7: Recall and precision at different thresholds for varying values of $\tau_a$ on the UCF-Crime dataset.

| $\tau_a$ | Recall | | | | | | | | | Precision | | | | | | | | | AP |
|---|---|---|---|---|---|---|---|---|---|---|---|---|---|---|---|---|---|---|---|
| | 0.1 | 0.2 | 0.3 | 0.4 | 0.5 | 0.6 | 0.7 | 0.8 | 0.9 | 0.1 | 0.2 | 0.3 | 0.4 | 0.5 | 0.6 | 0.7 | 0.8 | 0.9 | |
| 0.99 | 76.1 | 67.7 | 58.4 | 50.8 | 44.6 | 39.8 | 33.3 | 28.4 | 23.1 | 22.1 | 24.7 | 27.0 | 28.6 | 30.0 | 31.9 | 32.4 | 33.5 | 35.7 | 32.8 |
| 0.98 | 84.9 | 77.7 | 71.4 | 63.8 | 56.4 | 47.8 | 42.8 | 35.9 | 29.7 | 19.9 | 22.1 | 24.0 | 25.8 | 27.6 | 28.6 | 30.0 | 30.7 | 33.0 | 33.1 |
| 0.97 | 79.6 | 70.9 | 61.1 | 51.2 | 44.0 | 36.5 | 24.5 | 19.8 | 15.3 | 21.3 | 24.4 | 26.8 | 28.8 | 31.3 | 33.8 | 35.0 | 40.4 | 50.6 | 34.5 |
| 0.96 | 82.7 | 76.9 | 70.3 | 68.8 | 56.7 | 47.9 | 40.9 | 34.8 | 29.0 | 20.0 | 22.7 | 24.6 | 26.0 | 28.0 | 29.1 | 30.6 | 32.3 | 35.5 | **34.7** |
| 0.95 | 84.2 | 76.6 | 70.0 | 63.2 | 56.6 | 49.8 | 43.5 | 36.7 | 29.6 | 20.0 | 22.0 | 23.7 | 24.9 | 26.2 | 28.2 | 29.5 | 30.8 | 32.8 | 34.6 |
| 0.94 | 86.8 | 80.0 | 74.3 | 67.1 | 58.2 | 51.6 | 45.0 | 37.3 | 31.6 | 19.4 | 21.6 | 23.1 | 24.7 | 26.1 | 27.7 | 29.2 | 30.2 | 33.8 | 34.2 |

Table 8: Recall and precision at different thresholds for varying values of $\tau_a$ on the XD-Violence dataset.

| $\tau_a$ | Recall | | | | | | | | | Precision | | | | | | | | | AP |
|---|---|---|---|---|---|---|---|---|---|---|---|---|---|---|---|---|---|---|---|
| | 0.1 | 0.2 | 0.3 | 0.4 | 0.5 | 0.6 | 0.7 | 0.8 | 0.9 | 0.1 | 0.2 | 0.3 | 0.4 | 0.5 | 0.6 | 0.7 | 0.8 | 0.9 | |
| 0.99 | 84.3 | 78.7 | 73.9 | 68.1 | 61.9 | 54.7 | 48.2 | 41.6 | 34.1 | 64.8 | 68.3 | 71.5 | 74.6 | 77.2 | 79.9 | 82.3 | 84.8 | 87.7 | 78.68 |
| 0.98 | 87.5 | 82.4 | 76.8 | 70.3 | 62.9 | 55.7 | 47.5 | 38.7 | 29.6 | 62.7 | 66.7 | 70.0 | 73.3 | 76.3 | 79.7 | 82.9 | 86.8 | 89.8 | 79.13 |
| 0.97 | 87.6 | 84.2 | 80.0 | 76.0 | 71.0 | 65.3 | 59.1 | 53.8 | 47.3 | 62.0 | 65.4 | 67.6 | 70.0 | 72.1 | 74.5 | 76.5 | 78.8 | 81.5 | 78.81 |
| 0.96 | 87.5 | 83.5 | 79.1 | 75.0 | 69.6 | 64.0 | 56.9 | 51.1 | 43.8 | 61.8 | 65.2 | 67.9 | 70.4 | 73.2 | 75.9 | 78.7 | 81.0 | 83.9 | 79.32 |
| 0.95 | 90.5 | 87.2 | 83.7 | 79.9 | 75.2 | 70.2 | 63.4 | 57.4 | 50.0 | 60.0 | 63.4 | 66.2 | 68.7 | 71.4 | 74.1 | 77.3 | 79.8 | 82.7 | **80.88** |
| 0.94 | 92.1 | 88.8 | 85.0 | 81.0 | 75.6 | 96.9 | 63.1 | 56.5 | 47.8 | 56.6 | 60.5 | 63.5 | 66.4 | 69.0 | 72.0 | 75.4 | 78.8 | 82.5 | 79.29 |

## E.2  EFFECT OF $\tau_u$

In Table 10 and 9, we study the effect of varying the uncertainty threshold $\tau_u$. Note that uncertainty is the standard deviation of the model's predicted score over an ensemble of 3 models. A low threshold ($\tau_u = 0.10/0.15$) would render most predictions as uncertain, thereby enabling the model to explore all such events, causing overexploration. This would not prioritize the truly uncertain cases with high standard deviation, and thus lead to a low AP score. Similarly, a high threshold would result in an under-exploration. The tables show that a good balance occurs around $\tau_u = 0.20/0.25$ for UCF-Crime, and around $\tau_u = 0.30/0.35$ for XD-Violence datasets.

Table 9: Recall and precision at different thresholds for varying values of $\tau_u$ on the UCF-Crime dataset.

| $\tau_u$ | Recall | | | | | | | | | Precision | | | | | | | | | AP |
|---|---|---|---|---|---|---|---|---|---|---|---|---|---|---|---|---|---|---|---|
| | 0.1 | 0.2 | 0.3 | 0.4 | 0.5 | 0.6 | 0.7 | 0.8 | 0.9 | 0.1 | 0.2 | 0.3 | 0.4 | 0.5 | 0.6 | 0.7 | 0.8 | 0.9 | |
| 0.10 | 83.7 | 76.7 | 69.7 | 63.8 | 57.5 | 49.0 | 41.8 | 35.3 | 29.5 | 21.0 | 23.3 | 24.6 | 26.4 | 28.5 | 29.3 | 30.2 | 31.1 | 34.7 | 35.78 |
| 0.15 | 83.0 | 75.9 | 70.0 | 61.1 | 54.1 | 48.6 | 42.0 | 35.9 | 29.5 | 20.3 | 22.8 | 25.3 | 26.2 | 27.8 | 29.8 | 31.0 | 32.3 | 34.0 | 35.20 |
| 0.20 | 88.4 | 81.9 | 73.4 | 67.2 | 61.9 | 55.1 | 48.3 | 39.8 | 31.6 | 18.2 | 20.7 | 22.2 | 24.0 | 26.3 | 28.5 | 31.0 | 32.8 | 36.3 | 36.42 |
| 0.25 | 84.4 | 79.1 | 72.3 | 64.7 | 59.3 | 51.8 | 45.1 | 39.3 | 32.1 | 19.7 | 22.5 | 24.4 | 25.8 | 27.9 | 29.6 | 30.8 | 32.9 | 34.9 | **36.47** |
| 0.30 | 85.9 | 79.4 | 71.5 | 65.2 | 59.3 | 53.4 | 44.8 | 36.6 | 29.2 | 19.6 | 22.0 | 23.6 | 25.3 | 27.2 | 29.3 | 31.2 | 32.8 | 36.7 | 35.92 |
| 0.35 | 84.7 | 79.8 | 72.7 | 66.1 | 59.5 | 52.0 | 44.7 | 38.7 | 31.5 | 19.7 | 22.2 | 23.5 | 24.6 | 26.8 | 28.2 | 29.7 | 31.3 | 33.2 | 35.39 |
| 0.40 | 83.4 | 75.3 | 67.2 | 61.8 | 54.8 | 47.8 | 41.6 | 36.3 | 29.6 | 20.4 | 22.6 | 24.0 | 26.6 | 28.6 | 29.9 | 30.7 | 32.2 | 34.8 | 35.00 |

## E.3  SENSITIVITY TO PROMPT

To identify the best-performing prompt, we experiment with different prompting settings (Table 11) using only abnormal videos from the test set of the UCF-Crime dataset. Since the evaluation is performed on abnormal videos, we report AUROC as a preliminary metric for these experiments, as it is less inflated than when normal videos are included. Binary prompt with no anomaly prior indicates that VLM should generate only 1 or 0 output without any context of abnormal events. The only difference in the 'Definition' prompt is the removal of the specific anomaly type, while in the

Table 10: Recall and precision at different thresholds for varying values of $\tau_u$ on the XD-Violence dataset.

| $\tau_u$ | Recall | | | | | | | | | Precision | | | | | | | | | AP |
|---|---|---|---|---|---|---|---|---|---|---|---|---|---|---|---|---|---|---|---|
| | 0.1 | 0.2 | 0.3 | 0.4 | 0.5 | 0.6 | 0.7 | 0.8 | 0.9 | 0.1 | 0.2 | 0.3 | 0.4 | 0.5 | 0.6 | 0.7 | 0.8 | 0.9 | |
| 0.10 | 91.5 | 88.1 | 84.3 | 80.4 | 75.1 | 69.6 | 62.2 | 55.3 | 46.6 | 58.9 | 62.0 | 64.8 | 67.3 | 69.9 | 73.0 | 75.7 | 78.4 | 81.3 | 78.70 |
| 0.15 | 92.2 | 89.1 | 85.8 | 82.5 | 77.90 | 72.6 | 65.7 | 60.0 | 52.1 | 57.3 | 60.4 | 63.4 | 65.9 | 68.8 | 71.1 | 74.3 | 77.1 | 80.2 | 79.70 |
| 0.20 | 94.1 | 91.8 | 88.3 | 85.1 | 80.6 | 75.7 | 69.0 | 62.7 | 54.4 | 54.4 | 58.1 | 61.2 | 64.1 | 66.8 | 69.6 | 72.7 | 76.2 | 79.9 | 79.13 |
| 0.25 | 91.8 | 88.6 | 85.4 | 81.7 | 76.6 | 71.5 | 64.3 | 58.1 | 49.8 | 56.5 | 60.2 | 63.2 | 65.7 | 68.5 | 71.1 | 74.2 | 77.8 | 81.6 | 78.95 |
| 0.30 | 90.5 | 87.2 | 83.7 | 79.9 | 75.2 | 70.2 | 63.4 | 57.4 | 50.0 | 60.0 | 63.4 | 66.2 | 68.7 | 71.4 | 74.1 | 77.3 | 79.8 | 82.7 | **80.88** |
| 0.35 | 60.1 | 86.3 | 82.1 | 78.1 | 72.8 | 67.3 | 60.2 | 54.5 | 47.0 | 59.1 | 62.7 | 65.9 | 68.7 | 71.7 | 74.8 | 77.9 | 80.6 | 83.7 | 79.57 |
| 0.40 | 90.4 | 86.2 | 81.8 | 77.3 | 71.7 | 65.7 | 58.9 | 52.2 | 43.4 | 59.3 | 63.4 | 66.4 | 69.2 | 72.1 | 74.8 | 77.9 | 81.1 | 84.0 | 79.53 |

'Likelihood' prompt, the output is a value between 0 and 100. Since we cannot leverage the exact class type of abnormal event during training, we select a likelihood definition prompt to compare against different models.

**Prompt P1: Binary No Anomaly Prior**

```
You are an anomaly detection assistant.
Analyze the video clip carefully.
If the video clip contains any abnormal activity, reply with only "1".
If the video clip is normal, reply with only "0".
Do not include any explanation or extra text.
```

**Prompt P2: Binary Anomaly Prior**

```
You are a video anomaly detection analyst.

You are given a video clip and your task is to detect whether it
    contains any abnormal events like: [{anomaly}].

Definitions:
- Abnormal Event: Intentional, harmful, unlawful, or dangerous
    activities that threaten safety, break laws,
  or strongly deviate from normal daily routines. Examples include
      abuse, arrest, arson, assault,
  road accidents, explosions, fighting, shooting, vandalism,
      shoplifting, robbery, burglary, and similar
  threatening behaviors.

- Normal Event: Routine and harmless daily activities such as
    walking, talking, driving normally,
  shopping, exercising, or working.

Evaluation Criteria:
- Focus specifically on whether the given clip contains the
    abnormal event type [{anomaly}].
- Consider human actions, interactions, objects, and context.

Output Rule:
- If the clip shows [{anomaly}]     Reply with: 1
- If the clip does not show [{anomaly}]     Reply with: 0
- Reply with only the digit (0 or 1), no explanations or extra
    text.
```

E.4 EFFECT OF TEMPORAL WINDOW AND FRAME SELECTION

As shown in Table 12, we observe only minimal performance differences across different temporal windows and frame selection strategies. Therefore, we adopt the smallest temporal window, which provides a more fine-grained frame-level analysis without sacrificing overall performance.

Table 11: Sensitivity to Prompt.

| Model | Prompt Output | Anomaly Prior | AUROC (Abnormal Video) |
|---|---|---|---|
| InternVL3-14B | Binary | No | 68.67 |
| InternVL3-14B | Binary | Defination | 70.00 |
| InternVL3-14B | Binary | Anomaly Prior | 71.66 |
| InternVL3-14B | Binary | Reasoning and Anomaly | 73.03 |
| InternVL3-14B | Likelihood | Defination | 70.93 |

Table 12: Effect of Temporal Window and Frame Selection on InternVL.

| Frame Selection | Temporal Window | Number of Frames | AUROC (Abnormal Video) |
|---|---|---|---|
| Uniform | 48 | 8 | 70.00 |
| Uniform | 144 | 8 | 70.84 |
| Uniform | 288 | 8 | 70.78 |
| Uniform | 144 | 16 | 70.54 |
| Uniform | 288 | 16 | 70.62 |
| FPS | 288 | 8 | 70.48 |

## F    ABLATION ON VLMS

IN the table 13, we compared the effectiveness of a recent SOTA multimodal large language model trained on a video dataset using the "Detailed Likelihood" prompt with a temporal window of 48 segments and 8 frame selections per window. Constrained by inference resources, we select InternVL3-14B for our knowledge fusion module.

Table 13: Evaluation of different VLMs.

| Model | Model Size | AUROC (Abnormal) | AUROC Overall |
|---|---|---|---|
| InternVL3-9B | 9B | 66.89 | 73.82 |
| InternVL3-14B | 14B | 69.60 | 80.28 |
| InternVL3_5-14B | 14B | 69.59 | 77.97 |
| LLaVA-NeXT-Video | 7B | 66.11 | 74.9 |
| Vera (Ye et al., 2025) VLM Output | – | 62.46 | 75.26 |

## G    RESULT ON XD-VIOLENCE

In Figure 5, we analyze results for the XD-Violence dataset. We observe that our method achieves higher recall with minimal compromise in precision, similar to the UCF-Crime dataset as described in the main paper. We also observe that the misclassification cost for our method is lower than the UR-DMU baseline. The lower recall at 0.9 threshold is due to temporal clustering-based selection (TCBS).

## H    ABLATION ON DUAL EXPLORATION

We study the effect of each exploration strategy of our method, i.e., TCBS and UBS, in Figure 6. We observe that both TCBS and UBS alone can improve recall at a reasonable threshold of 0.4-0.7. Moreover, combining both achieves the highest recall along the threshold range with minimal compromise in precision.

## I    QUANTITATIVE ANALYSIS

Figure 7 illustrates cases where our method detects abnormal events more effectively than the baseline. The first two examples, Figures 7a and 7b, show videos where both models identify them as abnormal; however, the baseline method fails to capture all abnormal events that occur at distinct times. In contrast, our dual exploration strategy successfully explores other segments in the

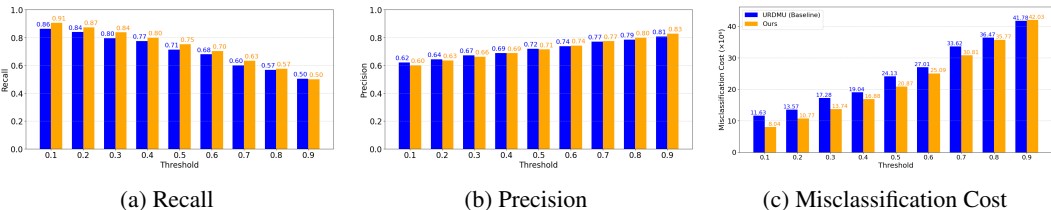

(a) Recall

(b) Precision

(c) Misclassification Cost

Figure 5: Comparison of recall, precision, and misclassification cost with the baseline (UR-DMU) across different thresholds on the XD-Violence dataset.

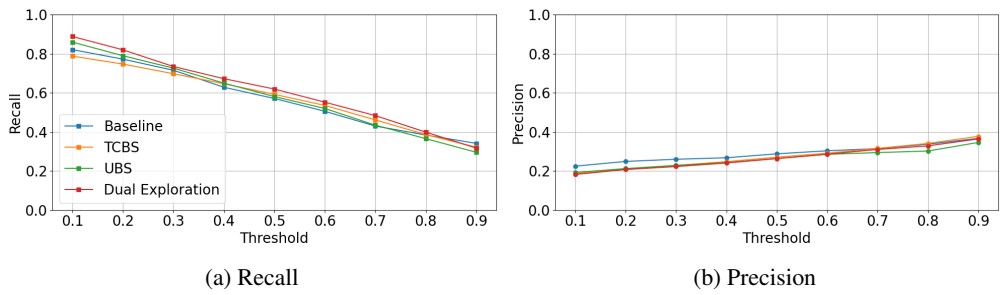

(a) Recall

(b) Precision

Figure 6: Comparison of Recall and Precision across thresholds for different ablation settings on UCF-Crime Dataset.

video. Moreover, as shown in Figures 7c and 7d, our diverse exploration can detect abnormal videos entirely missed by the baseline. Figures 7e to 7h present normal video cases where the baseline misclassifies them as abnormal.

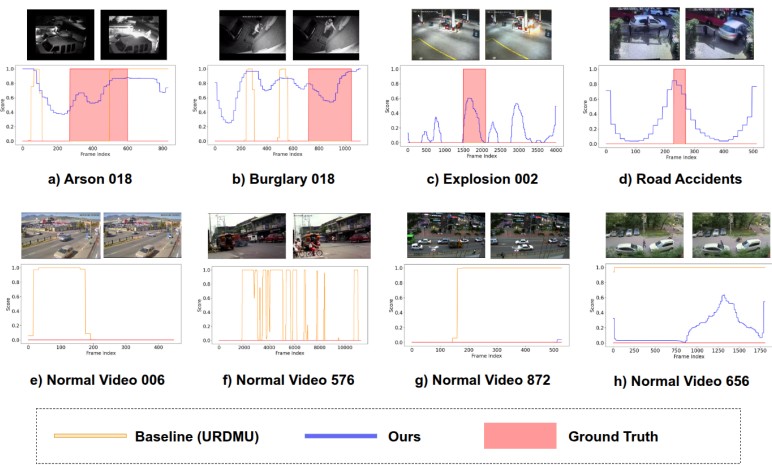

Figure 7: Quantitative analysis of baseline (UR-DMU) and our method on the UCF-Crime dataset.

## J    FIGURE REFERENCE FOR MAIN PAPER

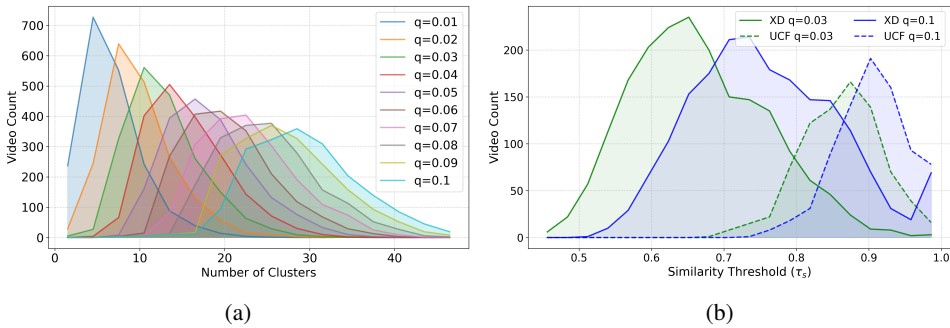

(a)                              (b)

Figure 8: (a) Cluster distribution for varying $q$ in XD-Violence. (b) Similarity threshold ($\tau_s$) distribution for varying $q$.

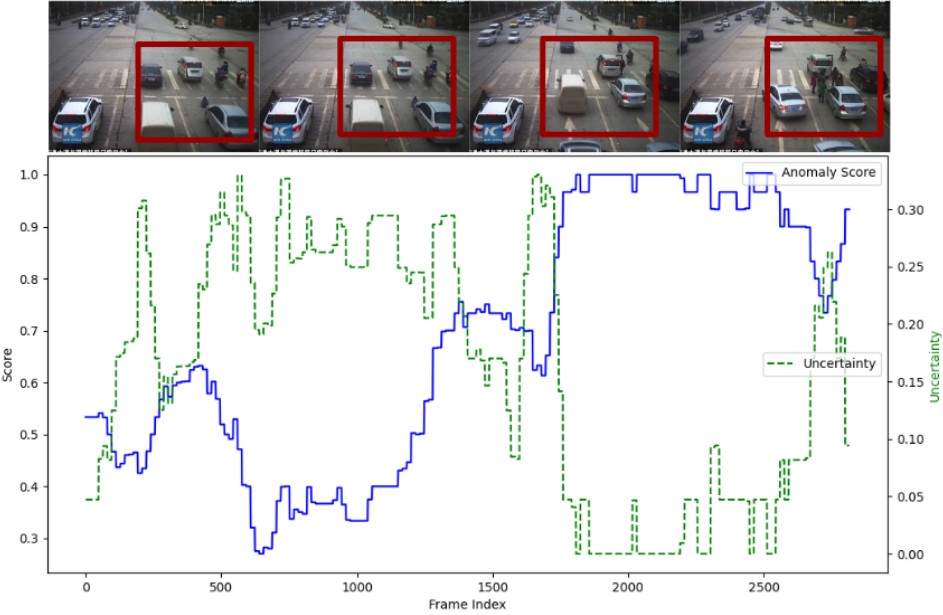

Figure 9: Top: Frames sampled from an abuse video where a person is repeatedly hit by a car, depicting subtle abnormal events. These events receive low prediction scores (blue line in the bottom graph) but exhibit high prediction uncertainty (green line). The model detects drastic motion changes, reflected by high anomaly scores with low uncertainty, after a group of people gathers near the scene, but it misses the actual abnormal event.

