# OpenReview forum: "Diversity–Ambiguity Exploration for Weakly Supervised Video Anomaly Detection"
_ICLR.cc/2026/Conference — ICLR 2026 Conference Withdrawn Submission_

### Official Review · Reviewer_5DoT · 2025-10-16

**Soundness:** 3
**Presentation:** 2
**Contribution:** 2
**Rating:** 2
**Confidence:** 5

**Summary:**

Diversity–Ambiguity Exploration for Weakly Supervised Video Anomaly Detection” proposes a novel dual exploration framework for weakly supervised video anomaly detection (WSVAD). The method combines temporal clustering to ensure diversity among segments and uncertainty-based selection to focus on ambiguous samples, improving recall and reducing false negatives compared to traditional top-k approaches. The authors also critique AUROC as an over-optimistic metric and recommend more practical measures like Recall@FPR and Average Precision.

**Strengths:**

The authors highlight a common issue in video anomaly detection (VAD) research—the inadequacy of the AUC metric for practical evaluation. I agree with the authors’ observation and appreciate their effort to raise awareness of this limitation. Overall, the proposed method is technically sound.

**Weaknesses:**

The discussion of AUC’s failure is somewhat redundant and would benefit from stronger experimental evidence. Several classical baselines and widely used VAD methods are missing from the comparative analysis, making it difficult to fully validate the claimed shortcomings of AUC.
Moreover, the use of AP and Recall@FPR as alternative metrics is not novel—these metrics have been adopted in previous studies. As a result, the overall originality of the paper appears limited.

**Questions:**

I recommend that the authors include more comprehensive comparisons with established VAD methods and explicitly analyze the differences between AUC, AP, and Recall@FPR in the revision. Such an expanded evaluation would more convincingly demonstrate the limitations of AUC and the practical advantages of alternative metrics.

Additionally, the paper’s contributions feel somewhat dispersed—one part focuses on the evaluation metric critique, while another introduces an uncertainty-based dual exploration method. Since these two aspects are only loosely connected, I suggest considering a clearer integration strategy or dividing them into two separate studies for better focus and impact.

---

### Official Review · Reviewer_C7Cu · 2025-10-25

**Soundness:** 1
**Presentation:** 1
**Contribution:** 1
**Rating:** 2
**Confidence:** 4

**Summary:**

A video anomaly detection method is proposed which is based on some existing methods such as URDMU. The proposed approach is memory based and uses multiple instance learning. Clustering is also employed to improve the anomaly detection performance.

**Strengths:**

Good performance is reported on two commonly used datasets.

**Weaknesses:**

1. First contribution of this work is "We identify a critical weakness in using AUROC for performance evaluation". There is no analysis presented in the paper to support this claim. Fig. 1 shows frame count on an unknown dataset but referred to as segments in the title and text. 15K frames and 15K segments are not equal. Just mentioning that 15k frames are missed does not prove or disprove any thing, given that the total number of frames is unknown. Make your analysis thorough.

2. There is no modification in the proposed method to handle the identified weakness in AUROC. How your method improves over this weakness you identified?

3. There is no better proposed measure found in the paper. Is the Area Under the Precision- Recall Curve free from the weakness identified in AUROC? Give thorough analysis that AUPRC is not suffering from the same weakness as AUROC.

4. Also, authors should consider other measures such as F1 score or EER commonly used by VAD methods. Are these measures also suffer from the same weakness as AUROC?

5. What is novel in the proposed method? It looks just a mix of existing approaches. Authors need to show the novelty of their approach. Method looks heavily based on URDMU: " Dual memory units with uncertainty regulation for weakly supervised video anomaly detection".

6. The abstract of the paper and many other sections are not specific rather look like a broad overview similar to the content generated by LLMs.

**Questions:**

Please answer the questions raised in the weaknesses section.

**Details Of Ethics Concerns:**

No ethical concern.

---

### Official Review · Reviewer_5wUs · 2025-10-31

**Soundness:** 2
**Presentation:** 3
**Contribution:** 2
**Rating:** 4
**Confidence:** 5

**Summary:**

This paper presents a dual-exploration approach to the task of Weakly-Supervised Video Anomaly Detection (WSVAD) along with arguing for improved metrics to evaluate such models. To ensure diversity in the video segments selected for the MIL loss, the authors propose to select the segments from abnormal videos via a clustering and thresholding approach, named TCBS. Segments of abnormal videos are clustered together if the cosine similarity between a segment and the following one exceeds threshold, which is computed for each abnormal video in the training set. From the resulting clusters, the segment with the highest anomaly score is selected for the MIL loss only if its anomaly score exceeds a predefined threshold. The second exploration strategy leverages the uncertainty quantified from the predictions of an ensemble of models and a memory unit. They apply cross-entropy loss between the segments of an abnormal video that satisfy two conditions:
* computed uncertainty between an ensemble of models is higher than threshold
* cosine similarity between the segment and the memory unit is higher than threshold

The final anomaly score is given by a weighted sum of the anomaly score predicted by the baseline model and the anomaly score assigned by a VLM. The authors propose to evaluate the WSVAD task using the Area Under Precision-Recall Curve (AP) and Recall @ False Positive Rates (FPR) metrics.

**Strengths:**

* The writing is clear and easy to follow, the figures are informative.
* The arguments for replacing the AUROC metric as the standard metric for evaluating WSVAD tasks are compelling and well posed.
* The proposed approach is well thought out and presented

**Weaknesses:**

The methodology presented in this paper relies on a large number of hyperparameters, thresholds and fine-tuned settings. This is not necessarily a downside to the presented methodology. However, it requires a more comprehensive and precise experimental evaluation than the one presented in the main paper to be correctly assessed. The following are observations based on the main paper text.

* In lines 108-109, the authors indicate the design of a anomaly-specific memory unity as one of the main contributions of the paper. However, the design of such memory unit is not described in the paper, while only its usage is described in Section 3.5.
* In lines 328-331, the authors acknowledge that the uncertainty threshold could be controlled by a scheduler, however they decide to manually set it to obtain a similar effect. The ablation studies of all hyperparameters are only shown in Appendix E.
* Table 1 and 2 compare the proposed approach with existing methods using as metrics AP and recall at three different false positive rates. While in some cases the authors report all four metrics, for most methods they do not report the recall results obtained by their own implementations of existing methods.
* Table 2 presents results on the XD-Violence dataset. While the authors compare their method with PEL4VAD on UCF-Crimes in Table 1, they do not include it in Table 2. In the PEL4VAD paper, the authors report an AP score of 85.59\%, which is 1\% higher than the score obtained approach proposed in this paper.

The main paper does not include any ablation studies on the hyperparameters, which are included in the Appendix. Given the importance of such hyperparameters to the proposed approach, the following are observations on the Appendix text.

* This approach relies on many fine-tuned hyperparameters, obtained by a very thorough grid search. Six hyperparameters are dataset-specific (Appendix C, lines 708-711), while others seem to be task-specific (Appendix E.3 and Appendix E.4, prompt engineering and frames selection for the VLM component). Tables 7, 8, 9 and 10 show quite high sensitivity to small changes for some of them (i.e. Table 7 and Table 8).
* The choice of the VLM component seems to be based on a comparison between VLM measured via the AUROC metric (Appendix F, Table 13). It seems counterintuitive to use AUROC to make this comparison given the emphasis placed on the inadequacy of AUROC throughout the paper (Abstract, lines 0.15-020; Introduction, lines 053-070 and 104-105; Section 3.2). While the authors mention that they choose InternVL3-14B due to resources constraints, they do not ablate the performance of the method with smaller VLMs such as InternVL3-9B that obtain a lower AUROC score.

**Questions:**

* What is the design of the anomaly memory unit?
* The paper reports the average number of clusters resulting from TCBS, but it is unclear how many segments are usually selected (at most one segment per cluster (line 269), if above a threshold (Eq. 4)). How many anomalous segments are usually selected for the MIL loss? If there are no segments above the threshold, is the video skipped entirely?
* Is the amount of segments selected from normal videos fixed or is it adjusted according to how many are selected from the anomaly videos via TCBS?

---

### Official Review · Reviewer_rKCg · 2025-10-31

**Soundness:** 2
**Presentation:** 3
**Contribution:** 2
**Rating:** 4
**Confidence:** 5

**Summary:**

Key Idea : The paper addresses a limitation in Weakly Supervised Video Anomaly Detection approaches: a contradiction between high performance (high Area Under the Receiver Operating Characteristic Curve (AUROC)) and low real world utility (poor Recall score at a meaningful False Positive Rate). The authors assume that this phenomenon stems from the model’s inability to capture all the diverse and rare anomalies due to weak video-level annotation and data imbalance. To tackle this problem, the paper proposes a dual exploration framework with a novel MIL (Multiple Instance Learning) approach. Exploring Diversity with a Temporal Clustering-based Selection (TCBS) and exploring Ambiguity with a Uncertainty-based Selection (UBS). By selecting a more informative set of segments for training, the paper claims to enhance the robustness of the model and to improve the performances in VAD.

Results : Experiments are conducted on two datasets, XD-Violence and UCF-Crime to validate the proposed method.

**Strengths:**

- Identifies a crucial problem : The paper highlights the problem of AUROC inadequacy as being the only metric for a WSVAD study, considering a model can have a high AUROC score and a low recall showing a high number of false negatives in the model predictions. In real-world scenarios like security or medical diagnosis, the robustness of a model is an important benefit.

- Novel dual exploration strategy : The major contribution of the paper is the combination of the two segment selection methods. TCBS addresses the challenge of providing the model with diverse anomalous patterns allowing it to encounter a wide variety of anomalies and not just the most common ones. UBS selects the high uncertainty segments making the model more robust to subtle and ambiguous anomalies.

**Weaknesses:**

- Limited novelty (Combination of known components):

The proposed method can be seen as a combination of following three techniques which makes the novelty incremental.

-- Temporal diversity via clustering is already explored in Bayesian non-parametric submodular video partition [a] (temporally aware clustering/partition, then select top items per cluster); the paper’s TCBS is a simpler, non-learned variant relying on adjacent-similarity thresholds and percentile heuristics. The conceptual goal, avoid selecting many adjacent high-score frames, matches prior art; what’s added is mostly a different thresholding recipe.

-- Uncertainty + memory guidance is a small variation on UR-DMU (dual memory, uncertainty regulation). Here, uncertainty is estimated by a tiny ensemble and filtered through cosine similarity to a memory bank. This preserves the same paradigm (uncertainty-aware memory mining) without a new learning principle or objective.

-- VLM fusion via late score averaging (Eq. 7) with a constant λ—no joint training, no rational weighting by confidence calibration, no prompt adaptation or distillation. This is standard ensembling, not a new integration strategy.

Overall, the paper reads as a sensible engineering combination of prior methods rather than a new algorithmic contribution with principled theory or novel modeling.

[a] SAPKOTA, Hitesh and YU, Qi. Bayesian nonparametric submodular video partition for robust anomaly detection. In: Proceedings of the IEEE/CVF conference on computer vision and pattern recognition . 2022. p. 3212-3221.

- Lack of comparison with state of the art performances in the Experiments section:

While being cited in the introduction as state of the art method for WSVAD, the performances of the PIVAD [b] paper are not in the results table. While this decision is explained in the supplementary material, it could have been interesting to add it to the table with a special mention or to explain this decision directly in the main paper.

[b] Majhi, S., D'Amicantonio, G., Dantcheva, A., Kong, Q., Garattoni, L., Francesca, G., ... & Brémond, F. (2025). Just Dance with pi! A Poly-modal Inductor for Weakly-supervised Video Anomaly Detection. In Proceedings of the Computer Vision and Pattern Recognition Conference (pp. 24265-24274).

- Classical Use of Vision-Language Models:

The VLM fusion step (Eq. 7) is not an integration, but a late-score averaging with a scalar λ. There is no co-training, prompt optimization, or knowledge transfer, unlike recent works (e.g., VadCLIP, TPWNG).

- VLM “knowledge sharing” is overstated:

InternVL is used training-free; fusion is a static convex combination. The experiments even show VLM excels on some categories (e.g., Explosion, Arrest) and underperforms on human-centric/short events (e.g., Abuse, Shooting), but the paper does not adapt prompts or weights per class, time scale, or confidence. Claims of “knowledge sharing” are therefore inflated.

- No mention of VLM utilisation in the main figure:

Considering the table 3 in the ablation studies section, the model performances are substantially benefiting from knowledge fusion with the VLM. An overview of the complete model with the VLM could provide information to the reader and enhance the paper readability.


- Many hyper parameters with various threshold:

The pipeline depends on several hand-tuned thresholds (q, τₛ, τₐ, τᵤ, τₘₛ, λ) picked via grid search per dataset; TCBS and UBS are not learned modules. This increases researcher degrees of freedom and risks overfitting to UCF-Crime/XD-Violence.

- Recall@FPR choice and operating thresholds:

the paper argues AUROC is misleading and uses Recall@FPR. This is unclear, and the selection of specific FPR values (1–3% for UCF-Crime, 4–6% for XD-Violence) and how they relate to operational constraints needs real-world justification. Also per-video vs global FPR is not clearly explained.

- Recomputed APs for baselines results may incur bias :

Re-implementations can introduce bias. The paper re-evaluates baselines to compute AP but does not detail re-training vs. re-scoring, nor verify fidelity to original implementations. It is also essential to compare proposed methods to the state-of-the-art methods in its original form of evaluation i.e. AUC and then also it is important to  see if the re-implementation of the same SoTA results in similar AUC. Otherwise, it is hard to judge the right potential of the proposed method.


- Computational cost:

The two exploration methods, diversity with the temporal clustering and ambiguity with the uncertainty estimation and sampling, add computational complexity to the training process. The paper could discuss the trade-off between performance gain and the training time and resource usage compared to other MIL methods for WSVAD.

- No cross-dataset generalization results:

The paper trains and evaluates models separately for each dataset, but there is no experiment showing cross-dataset transferability. Because the framework learns dataset-specific motion statistics, it might overfit to the type of anomalies in each dataset. Testing a model trained on UCF-Crime on XD-Violence would provide valuable insights into generalization and robustness.

- Key implementation details are incomplete or ambiguous:

exact prompt templates, whether VLM prompts used ground-truth anomaly labels at inference (risking label leakage).

- Event based evaluation Missing:

Further, the author should go for Event based evaluation such as (temporal IOU) to verify the robustness of the method.

**Questions:**

- Could you elaborate on the main contributions of your work and explain how your method does not consist of a combination of previous methods?

- Why did not you try to co-train, optimize prompt, or perform knowledge transfer using Vision-Language Models?

- Why is there no mention of VLM utilisation in the main figure?

- Since the False Positive Rate (FPR) is already used in computing AUROC, what is the rationale for preferring Average Precision (AP) over AUC? Typically, AUC evaluates the True Positive Rate (TPR) and FPR across multiple thresholds between 0 and 1, so it’s unclear what additional insight AP provides in relation to FPR.

- Why did you choose different specific parameters depending on the dataset you are testing on? (FPR 1-3% for UCF Crime, FPR 4-6% for XD-Violence)

---

### Note · Authors · 2025-11-14

I have read and agree with the venue's withdrawal policy on behalf of myself and my co-authors.